# KidSat: satellite imagery to map childhood poverty

## Abstract

Satellite imagery has emerged as an important tool to analyze demographic, health, and development indicators. While various deep learning models have been built for these tasks, each is specific to a particular problem, with few standard benchmarks available. We propose a new dataset pairing satellite imagery and high-quality survey data on child poverty to benchmark satellite feature representations. Our dataset consists of 33,608 images, each 10 km × 10 km, from 16 countries in Eastern and Southern Africa in the time period 1997-2022. As defined by UNICEF, multidimensional child poverty comprises six fundamental factors—housing, sanitation, water, nutrition, education, and health (UNICEF, 2021)—which can be calculated from geocoded, face-to-face Demographic and Health Surveys (DHS) Program data. Using our dataset we benchmark multiple feature representations for encoding satellite imagery, from low-level satellite imagery models such as MOSAIKS (Rolf et al., 2021), to deep learning foundation models, which include both generic vision models such as DINOv2 (Oquab et al., 2023) and specific satellite imagery models such as SatMAE (Cong et al., 2022). As part of the benchmark, we test spatial as well as temporal generalization, by testing on unseen locations, and on data beyond the training years. We provide open source code to reproduce and extend our entire pipeline: building the satellite imagery dataset, obtaining ground truth data from DHS, and comparing the various models considered in our work.

## 1 Introduction

Major satellites like those in the Landsat and Sentinel programs regularly circle the globe, providing updated, publicly available, high-resolution imagery every 1-2 weeks. An emerging literature in remote sensing and machine learning points to the promise that these large datasets, combined with deep learning methods, hold to enable applications in agriculture, health, development, and disaster response. A cross-disciplinary set of publications hint at the potential impact, showing how satellite imagery can be used to estimate the causal impact of electricity access on livelihoods (Ratledge et al., 2021), to measure income, overcrowding, and environmental deprivation in urban areas (Suel et al., 2021) and to predict the human population in the absence of census data (Wardrop et al., 2018). Despite these successes, machine learning for satellite imagery is not yet a well-developed field (Rolf et al., 2024), with current approaches overlooking the unique features of satellite images such as variation in spatial resolution over logarithmic scales (from < 1 meter to > 1 km) (Rolf et al., 2024) and the heterogeneous nature of satellite imagery in terms of the number of bands available from 3 bands for RGB to multispectral and hyperspectral.

Many areas of machine learning have advanced through the development of standardized datasets and benchmarks. Given the wide set of possible use cases for satellite imagery, there is no doubt room for multiple benchmarks. However there are only a few sources of up-to-date, high-quality satellite imagery, especially Landsat and Sentinel, so it is natural to construct publicly available datasets using these satellite programs. Given the proven effectiveness of remote sensing for tasks that are naturally visible from space, such as land usage prediction, crop yield forecasting, and deforestation, we instead choose to focus on a more difficult task: multidimensional child poverty estimation.

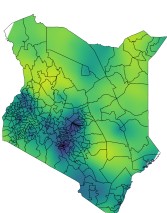
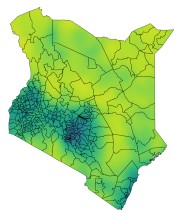
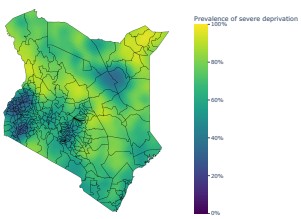
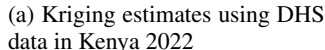

(a) Kriging estimates using DHS data in Kenya 2022

(b) DINOv2 fine-tuned on Kid-Sat spatial training dataset

(c) DINOv2 fine-tuned on KidSat temporal training dataset

Figure 1: Estimates of child poverty defined as the prevalence of severe deprivation for Kenya in 2022. (a) shows predictions using a spatial statistics approach, kriging on the cluster locations using Kenya DHS 2022 data only with a spherical variogram. (b) shows predictions from DINOv2 fine-tuned on the KidSat spatial dataset, in which 20% of all clusters in Eastern and Southern Africa are held out. (c) shows predictions from DINOv2 fine-tuned on the KidSat temporal dataset, in which the training data is from before 2020.

Of the 8 billion people in the world, over 2 billion are children (aged $< 18$ years old, as defined in the UN Convention on the Rights of the Child (UN General Assembly, 1989)). Child poverty is not the same as adult poverty; children are growing and developing so they have specific nutrition, health, and education needs—if these needs are not met, there can be lifelong negative consequences (Brooks-Gunn & Duncan, 1997). Poverty cannot be assessed simply by measuring overall household resources, as households may be highly unequal, and some of the needs of children, such as vaccines or education, may be neglected even in households that are not poor. Instead, child poverty must be measured at the level of the children and their experiences (UNICEF, 2021).

Child poverty is based on the "constitutive rights of poverty" (UNICEF, 2021). This means that child poverty encompasses key dimensions essential for children's well-being, such as education, health, and nutrition, which depend on material resources. However, it excludes non-material aspects, including neglect, violence, and lack of privacy. Crucially for the purposes of establishing a useful dataset and benchmark, the internationally agreed definition of child poverty was designed to enable cross-country comparisons (UNICEF, 2021).

In this work, we introduce a new dataset, KidSat, designed to provide a benchmark for applying advanced computer vision methods to the challenge of child poverty estimation. The dataset includes geocoded surveys from 16 countries across Eastern and Southern Africa, paired with multispectral satellite imagery. By offering this dataset and benchmark, our goal is to bridge state-of-the-art computer vision advances with real-world applications, addressing complex socioeconomic issues like child poverty through the lens of satellite imagery. Our contribution are as follows:

- We aggregate the geocoded DHS survey data on multidimensional child poverty alongside matching satellite imagery. This dataset is suitable for fine-tuning large models; we provide a univariate measure (the percentage of children experiencing severe deprivation, ranging from 0% to 100%), making model performance intuitively grasped by policymakers.

- We benchmark the performance of various models for these tasks, ranging from baseline models using spatial correlation to satellite-based foundational vision models. In particular, we demonstrate the importance of high imagery resolution and vision transformer architecture over CNNs for addressing this child poverty estimation task.

- We propose a fine-tuning approach for generating robust satellite feature representations, with the potential to be adapted for addressing challenges beyond child poverty estimation.

## 2 RELATED WORK

### 2.1 EXISTING SATELLITE IMAGERY DATASETS

With increased access to publicly available high-resolution satellite imagery through the Landsat and Sentinel programs, satellite imagery datasets have become very popular for training machine

learning models. Models and datasets include functional map of the world (fMoW) (Christie et al., 2018), XView (Lam et al., 2018), Spacenet (Van Etten et al., 2018), and Floodnet (Rahnemoonfar et al., 2021) where the tasks are object detection, instance segmentation, and semantic segmentation. These are computer vision-specific tasks, rather than applied health and economic prediction problems, meaning the use of these datasets and models may be inappropriate for applied health and development researchers and practitioners.

## 2.2 SATELLITE IMAGERY FOR DEMOGRAPHIC AND HEALTH INDICATORS

Machine learning models applied to satellite images are becoming more commonplace for analyzing demographic, health, and development indicators as they can increase coverage by allowing for interpolation and faster analysis in under-surveyed regions. In an early work, Jean et al. (2016) leveraged convolutional neural networks (CNNs) to process satellite images for tracking human development at increasing spatial and temporal granularity. Since then, satellite images have been used to track development indicators which are clearly visible from space such as agriculture and deforestation patterns (Ball et al., 2022; Estes et al., 2022; Xu et al., 2024) but also more abstract quantities such as poverty levels (Ayush et al., 2021), health indicators (Daoud et al., 2023), and the human development index (Sherman et al., 2023).

## 2.3 FOUNDATION SATELLITE IMAGERY MODELS

As increasing volumes of data become available, and with progress in self-supervised learning (He et al., 2022; Caron et al., 2021), many foundation models are emerging. In computer vision, these large models are pre-trained with self-supervised learning on hundreds of millions of images, serving as a "foundation" from which they can be fine-tuned for specific tasks. Popular examples of this are SimCLR (Chen et al., 2020), CLIP (Radford et al., 2021), and DINO (Caron et al., 2021). Recently, foundation models have been trained for satellite imagery specifically on vast amounts of unlabelled satellite images. Examples of these are SatMAE (Cong et al., 2022) based on masked autoencoders (He et al., 2022), SatCLIP (Klemmer et al., 2023) based on CLIP (Radford et al., 2021), and DiffusionSat (Khanna et al., 2024) which is a diffusion model (Rombach et al., 2022) for generating satellite images. As it is not yet clear whether there is a benefit from training foundation models on more specific, but smaller datasets, we benchmark both generic foundation models for computer vision as well as satellite-specific foundation models.

## 3 KIDSAT DATASET

In this section, we introduce our unique dataset, named KidSat, which is derived from the DHS Program combining high-resolution satellite imagery with detailed numerical survey data focused on demographic and health-related aspects in Eastern and Southern Africa. This dataset leverages the rigorous survey methodologies from DHS to offer high-quality data on health and demographic indicators, complemented by satellite images of the surveyed locations. The rich information embedded in the satellite images enables the application of advanced deep learning methods to estimate key indicators in unsurveyed locations. We provide the details of dataset statistics in Appendix A.1.

## 3.1 DEMOGRAPHIC & HEALTH SURVEYS AND CHILD POVERTY

Dating back to 1984, the DHS Program[1] has conducted over 400 surveys in 90 countries, funded by the US Agency for International Development (USAID) and undertaken in partnership with country governments. These nationally representative cross-sectional household surveys, with very high response rates, provide up-to-date information on a wide range of demographic, health, and nutrition monitoring indicators. Sample sizes range between 5,000 and 30,000 households, and are collected using a stratified, two-stage cluster design, with randomly chosen enumeration areas (EAs) called "clusters" forming the sampling unit for the first stage. In each EA, a random sample of households is drawn from an updated list of households. DHS routinely collects geographic information in all surveyed countries. Cluster locations are released with random noise added to preserve anonymity, with this "jitter" being different for rural and urban EAs.

---

[1]http://www.dhsprogram.com

The DHS data include both continuous and categorical variables, each requiring a different approach for aggregation to ensure accurate ecological analysis at the cluster level. For continuous variables, we calculated the mean of all responses associated with a particular spatial coordinate. Min-max scaling was applied after aggregation to normalize the data, ensuring that all values were on a scale from 0 to 1. Categorical variables were processed using one-hot encoding, which converts categories into binary indicator variables. Similarly, the mean of these binary representations was computed for each category at each cluster location.

| Dimension | Unit of Analysis | Severe Deprivation Definition |
|---|---|---|
| Housing | Children under 17 years of age | Children living in a dwelling with five or more persons per sleeping room. |
| Sanitation | Children under 17 years of age | Children with no access to a toilet facility of any kind. |
| Water | Children under 17 years of age | Children with no access to water facilities of any kind. |
| Nutrition | Children under 5 years of age | Stunting (3 standard deviations below the international reference population). |
| Education | Children between 5-14 years of age | Children who have never been to school. |
| | Children between 15-17 years of age | Children who have not completed primary school. |
| Health | Children 12-35 months old | Children who did not receive immunization against measles nor any dose of DPT. |
| | Children 36-59 months old | Children with severe cough and fever who received no treatment of any kind. |
| | Children 15-17 years old | Unmet contraception needs. |

Table 1: Severe deprivation definitions by dimension and unit of analysis. Table adapted from UNICEF (2021).

Child poverty was assessed using a methodology formulated by UNICEF that evaluates child poverty across six dimensions: housing, water, sanitation, nutrition, health, and education (UNICEF, 2021). Each child was classified as severely deprived or not by the definition in Table 1. An overall classification of severe deprivation is made if the child experiences severe deprivation on any of the six dimensions. Our target quantity of interest, severe_deprivation, was calculated as the percentage of children experiencing severe deprivation within a cluster. The detailed usage of DHS variables and the statistics of the target variable can be found in the Appendix A.1.2.

## 3.2 SATELLITE IMAGES

This study utilizes high-resolution satellite imagery from two primary sources: Sentinel-2 and Landsat 5, 7, and 8. These satellite programs are chosen for their public accessibility, their specific advantages in computer vision applications, and their wide temporal coverage.

Figure 2 presents a heatmap showing the density of DHS survey cluster locations included in the KidSat dataset. The survey collection spans most of the Eastern and Southern African countries over a wide range of years. At each cluster location, we obtained a 10 km × 10 km satellite image using Google Earth Engine (GEE). Selection criteria for the imagery include the year of the survey and prioritization based on the least cloud cover within that year.

Both Sentinel-2 and Landsat series satellites include RGB bands, crucial for standard object recognition tasks in computer vision. Beyond the RGB spectrum, these satellites offer a rich assortment of additional spectral bands for advanced remote sensing analysis, such as estimating vegetation density and water bodies.

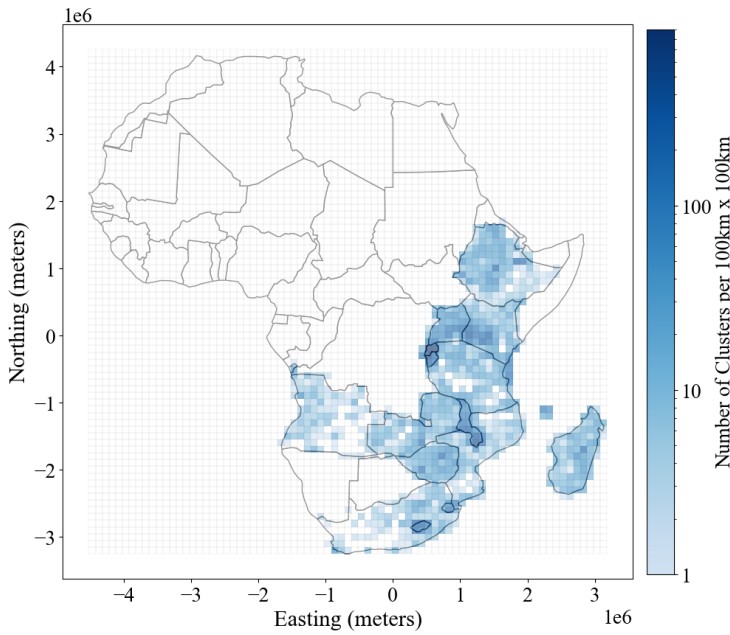

Figure 2: Heatmap showing the distribution of DHS cluster locations in Eastern and Southern Africa. The color represents the count of cluster locations per 100 km x 100 km grid cell.

## 4 BENCHMARK

### 4.1 SPATIAL

We use 5-fold spatial cross-validation at the cluster level across countries in Eastern and Southern Africa. We train our models on 80% of the clusters and evaluate their performances using the mean absolute error (MAE) of the `severe_deprivation` variable on the held-out 20% of clusters. This benchmark is designed to evaluate the model's capability to estimate poverty or deprivation levels at any given location based solely on satellite imagery data, quantifying the model's generalization capabilities to unsurveyed locations.

### 4.2 TEMPORAL

The temporal benchmark employs a historical data training approach, where we use data collected before 2019 (inclusive) as the training set to develop our models. The objective is to predict poverty in 2020 to 2022. Model performance is evaluated using the MAE of the `severe_deprivation` variable. This benchmark tests the model's ability to capture temporal trends and forecast poverty based on satellite imagery data. This capability is crucial for, e.g. nowcasting poverty before survey data becomes available.

### 4.3 MODELS TO BE COMPARED

We consider both baseline models and a range of more advanced computer vision models, both unsupervised and semi-supervised, with and without fine-tuning. Each model represents a distinct strategy for handling and processing satellite imagery:

**Baseline:** To assess baseline performance and demonstrate the improvements achieved by satellite-based methods, we employed two baseline approaches to benchmark the performance of traditional statistical models. Specifically, we utilized mean regression and Gaussian process regression as the baseline methods. In mean regression, the model simply predicts the average target value from the training set. The Gaussian process regression uses geo-coordinates as input and models the target child poverty variable by leveraging spatial proximity.

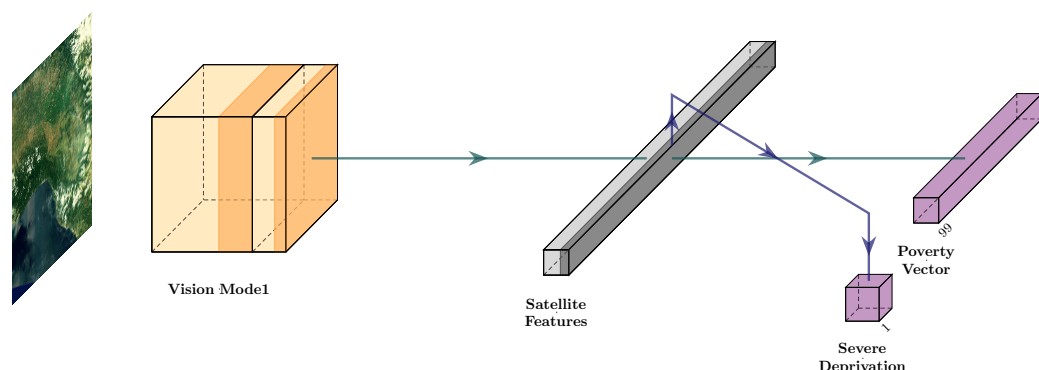

Figure 3: Illustration of the fine-tuning pipeline using foundational vision models: Satellite imagery is passed through the vision model to generate satellite features, which are then fine-tuned on a poverty vector. Evaluation is conducted using L2-regularized linear regression to predict the severe deprivation variable.

**MOSAIKS:** (Rolf et al., 2021) MOSAIKS is a generalizable feature extraction framework developed for environmental and socio-economic applications. We obtain MOSAIKS features from IDinsight, an open-source package that utilizes the Microsoft Planetary Computer API. The framework leverages satellite imagery to extract meaningful features from the Earth's surface. For our purposes, we used its Sentinel service, querying with specific coordinates, survey year, and a window size of 10 km × 10 km.

**DINOv2:** (Oquab et al., 2023) Initially designed for self-supervised learning from images, DINOv2 excels in generating effective representations from RGB bands alone. For our study, we selected the pre-trained base model with the vision transformer architecture as the backbone of our foundational model. We fine-tuned this foundational model with DHS variables to enhance its capability for predicting poverty. DINOv2 is evaluated in both its raw and fine-tuned forms using RGB imagery for both spatial and temporal benchmarks.

**SatMAE:** (Cong et al., 2022) SatMAE was originally developed for landmark recognition from satellite imagery. We extracted the encoder pipeline and fine-tuned it with DHS variables to enhance its performance for predicting poverty. SatMAE has 3 variants: RGB, RGB+temporal, and multispectral. For benchmarking, we used the RGB variant for the spatial benchmark, and RGB+temporal for the temporal benchmark. The RGB+temporal variant takes 3 images of different timestamps from the same location; however, to facilitate a direct comparison with the other methods which use only a single image, we provide SatMAE with the same image three times.

### 4.3.1 EVALUATION AND FINE-TUNING

In our fine-tuning pipeline shown in Figure 3, we start from DINOv2's and SatMAE's original checkpoints with an uninitialized head and train it against 17 selected DHS variables to minimize mean absolute error (MAE). We then evaluate the model by replacing the head with a cross-validated ridge regression model mapping satellite features to the severe_deprivation variable and calculate the MAE loss on a test set that was neither seen by the fine-tuned model nor the ridge regression. For the spatial task, we perform a 5-fold cross-validation on the whole dataset, and for the temporal benchmark, we take the training set as the data before the year 2020 and evaluate on the data from 2020 to 2022.

For DINOv2, we used a batch size of 8 for Landsat imagery and a batch size of 1 for Sentinel imagery, with L1 loss and an Adam optimizer of learning rate and weight decay both set to 1e-6. We trained the model for 20 epochs with Landsat imagery and 10 epochs with Sentinel imagery, selecting the model with the minimum validation loss on predicting the 17 DHS variables. Each task was trained on a single Nvidia V100 32GB GPU, with an average training time of 1 hour per epoch for Landsat and 2 hours per epoch for Sentinel imagery. For SatMAE, we resized the input to its pre-trained configuration ($224 \times 224$) and used a batch size of 32 for the spatial task and 16 for

Table 2: Comparison of MAE on `severe_deprivation` across benchmarks and imagery sources. In the spatial task, random clusters are held out, while the temporal task is a more difficult forecasting task, with the years 2020-2022 held out. While SatMAE is a foundation model trained with satellite imagery, it is outperformed by the more generic DINOv2.

| Model | Imagery Source | MAE $\pm$ SE (Spatial) | MAE (Temporal) |
|---|---|---|---|
| Mean Regression | - | $0.2930 \pm 0.0018$ | 0.3183 |
| Gaussian Process Regression | - | $0.2436 \pm 0.0002$ | 0.5656 |
| MOSAIKS | Sentinel-2 | $0.2356 \pm 0.0114$ | 0.2588 |
| DINOv2-ViT (Raw) | Landsat | $0.2260 \pm 0.0005$ | 0.2704 |
| DINOv2-ViT (Raw) | Sentinel-2 | $0.2013 \pm 0.0019$ | 0.2597 |
| DINOv2-ViT (Fine-tuned) | Landsat | $0.2042 \pm 0.0015$ | **0.2574** |
| DINOv2-ViT (Fine-tuned) | Sentinel-2 | $\mathbf{0.1873 \pm 0.0022}$ | 0.2858 |
| SatMAE (Raw) | Landsat | $0.2341 \pm 0.0017$ | 0.3453 |
| SatMAE (Raw) | Sentinel-2 | $0.2347 \pm 0.0027$ | 0.3067 |
| SatMAE (Fine-tuned) | Landsat | $0.2125 \pm 0.0019$ | 0.3376 |
| SatMAE (Fine-tuned) | Sentinel-2 | $0.2093 \pm 0.0039$ | 0.3139 |

the temporal task. The training was done with Adam optimizer with learning rate 1e-5 and weight decay 1e-6, for at most 20 epochs with the early stopping of patience 5 and delta 5e-4. Each task is trained on a single Nvidia L4 GPU, taking, for Landsat and Sentinel, 1 and 2 hours for the first epoch and 15 and 10 minutes for each subsequent epoch with data caching.

## 5 RESULTS

The performance of the child poverty prediction models is summarized in Table 2.

### 5.1 SPATIAL BENCHMARK

In the spatial benchmark, Gaussian process regression (GPR) with geographic coordinates resulted in a mean absolute error (MAE) that is 0.04 lower than that achieved by the baseline mean regression model. Notably, regressions using outputs from foundational vision models outperformed both the mean regression and GPR. The MOSAIKS features based on Sentinel-2 imagery achieve 0.2356 MAE on predicting the `severe_deprivation` variable. Utilizing Landsat imagery, the DINOv2 and SatMAE achieved MAEs of 0.2260 and 0.2341 respectively. Further enhancements through fine-tuning with DHS variables led to reduced prediction errors, with DINOv2 and SatMAE recording MAEs of 0.2042 and 0.2125 respectively. When using Sentinel-2 imagery, the SatMAE architecture achieved errors of 0.2347 and 0.2093 before and after the fine-tuning, while DINOv2 further lowered the errors to 0.2013 and 0.1873 respectively.

### 5.2 TEMPORAL BENCHMARK

In the temporal benchmark, models faced greater challenges in forecasting poverty. GPR was substantially worse than the mean regression. Using Sentinel-2 imagery, MOSAIKS recorded an MAE of 0.2588, with DINOv2 and SatMAE achieving MAEs of 0.2597 and 0.3067 respectively. Additional fine-tuning with DHS variables led to increased prediction errors, with DINOv2 and SatMAE resulting in MAEs of 0.2858 and 0.3139. Employing Landsat imagery, the pre-trained DINOv2 and SatMAE model achieved worse initial MAEs of 0.2704 and 0.3453; nevertheless, additional fine-tuning on DHS variables resulted in relative equal performance for both models, with MAEs of 0.2574 and 0.3376 respectively.

### 5.3 INTERPRETATION OF RESULTS

The performance of various child poverty prediction models is shown in Table 2. Our prediction task is the percentage of a location's children who are experiencing severe deprivation, so an MAE on the order of 0.20 is equivalent to 20 percentage points of error, which policymakers may consider not yet

low enough to be useful. The spatial benchmark demonstrates the advantage of using foundational vision models over the baseline mean prediction model and GPR. Models like MOSAIKS, DINOv2, and SatMAE, particularly when improved through fine-tuning with DHS variables, show a further reduction in mean absolute error. This implies that spatial features extracted from satellite imagery are comparably more effective than GP modeling in estimating poverty indicators in regions where surveys have not been conducted.

The temporal benchmark, which evaluates a forecasting task (predict 2020-2022 using data from before 2019), appears to be more difficult than the spatial benchmark. Satellite imagery is at best a proxy for multidimensional child poverty, and this finding suggests it is a better proxy for quantifying spatial as opposed to temporal variation. Satellite imagery models performed worse on the temporal as compared to the spatial benchmark, and the fine-tuned models, particularly those using Sentinel-2 imagery as the source input, showed increased MAE compared to the raw output from both DINOv2 and SatMAE models. This suggests that the models overfit the historical data, and struggled to generalize to data collected after 2020. GPR based on spatial coordinates had no way of predicting changes over time, explaining its very poor performance.

## 6 DISCUSSION

### 6.1 IMAGERY-MEMORY TRADE-OFF

As compared to Landsat, models utilizing Sentinel-2 imagery, such as the fine-tuned versions of DINOv2 and SatMAE, demonstrate improved performance in the spatial benchmarks. These models benefit from the high-resolution visible spectra provided by Sentinel-2, which enabled more precise predictions of deprivation levels across diverse geographical regions.

Additionally, the computational demands associated with processing high-resolution Sentinel-2 data present substantial challenges. For instance, large versions of vision transformers could not be accommodated within the memory constraints of a 32 GB GPU when processing the full Sentinel-2 data. In contrast, these larger models could be deployed with Landsat data, which offers lower resolution but requires less GPU memory. Under the spatial setting, this scenario highlights a critical trade-off in model deployment: the choice between employing lightweight models to retain the high resolution of Sentinel-2 imagery or opting for more powerful models that necessitate a reduction in image resolution to ensure feasibility.

### 6.2 MODELING COMPARISON

We consider a representative set of models: MOSAIKS is a basic statistical model, DINOv2 is a foundation model pre-trained on generic images, and SatMAE is a foundation model pre-trained on satellite imagery.

#### 6.2.1 MOSAIKS

MOSAIKS is designed to provide general-purpose satellite encodings and is notably accessible through Microsoft's Planetary Computer service. This model generates a large output vector, typically around 4000 dimensions, which, while comprehensive, can lead to increased computational costs when methods beyond simple linear regression are employed. Furthermore, although MOSAIKS is well-suited for broad applications, integrating online feature acquisition into a fine-tuning process tailored specifically to poverty prediction presents challenges. This limitation can hinder its effectiveness when adapting to specific tasks where dynamic feature updates are crucial. We also note that MOSIAKS' API at times returned no-features, even after implementing rate-limiting mechanisms. This random behaviour combined with unavailability of features before 2013 limits the use of MOSAIKS considerably.

#### 6.2.2 DINOv2

DINOv2 stands out as a state-of-the-art foundational model that excels in generating effective vector representations from RGB bands alone, achieving comparable performance to models that utilize additional spectral bands. Its flexibility in model sizing allows users to select the optimal model

scale for specific training needs, enhancing its utility across various computational settings. The availability of pre-trained weights simplifies the process of fine-tuning for specialized tasks such as poverty prediction. However, DINOv2's reliance solely on RGB bands means it does not leverage the broader spectral information available in other satellite imagery bands, which may limit its application scope to scenarios where such data could provide additional predictive insights.

### 6.2.3 SATMAE

SatMAE demonstrates respectable results, surpassing baseline models even with only its raw, pre-trained configuration. The pre-trained SatMAE model is configured to process images of $224 \times 224$ pixels, constraining its ability to utilize higher-resolution imagery, such as the $1000 \times 1000$ pixel images from Sentinel-2. This limitation restricts its performance, particularly in comparison to models that can fully exploit high-resolution data (e.g. DINOv2), thereby failing to match the effectiveness of other advanced models in our analysis. We also note in Appendix A.3.2 that resizing the input imagery to $224 \times 224$ would also decrease the DINOv2 performance, thereby again highlighting the importance of processing high-resolution imagery.

In this study, we aim for a direct comparison between DINOv2 and SatMAE's feature representations. Therefore, we provide both models only RGB satellite imagery as input for benchmark performance, evaluating the model's capability of capturing spatial and temporal patterns solely relying on the imagery. While SatMAE fails to match the performance of DINOv2, we note that SatMAE may have the potential for increased performance when benefiting from additional data such as temporal encoding and wider spectral information.

### 6.3 ADDITIONAL INVESTIGATION

#### 6.3.1 DIRECTION OPTIMIZATION ON THE TARGET VARIABLE

In this work, we present our fine-tuning scheme for foundational vision models as follows (illustrated in Figure 3): We pass the imagery through the vision model to generate feature representations of the satellite imagery and map the feature vector to a vector transformed from the 17 DHS variables, which are used to calculate the desired severe deprivation variable. The model's parameters are updated through back-propagation of the L1 loss on the poverty vector. It is natural to ask whether the results would be optimized if we directly fine-tuned using the target `severe_deprivation` variable. However, as the results shown in Appendix A.3.3 demonstrate, direct optimization on the single variable does not improve and may even worsen the performance of child poverty estimation. One explanation is that when fine-tuned on the higher-dimensional poverty vector, the satellite features generated become more robust; when solely optimizing the target variable, the model is more prone to overfitting, which undermines generalizability to unseen locations.

#### 6.3.2 MODEL ARCHITECTURE COMPARISON: CNN VS. VIT

Despite the emergence of Vision Transformer (ViT) architectures (Dosovitskiy et al., 2021), applications of satellite vision models in socio-economic research have predominantly utilized CNN architectures (Xie et al., 2016; Thirumaladevi et al., 2023; Jean et al., 2016), while models with ViT backbones have not been widely applied (Kumari & Kaul, 2023). To assess the effect of architectural change, we replicate the experiment using the DINO-ResNet50 model and present the results in A.3.1. Note that since DINO-ResNet50 is based on a CNN architecture, it requires a fixed input size of $224 \times 224$. We observe that DINO-ResNet50 fails to match the performance of SatMAE when using data of the same resolution. Furthermore, when compared to DINOv2 with a ViT backbone, the ResNet-based model exhibits a larger performance discrepancy, likely due to both lower satellite resolutions and architectural differences. This experiment highlights the improvements achieved by using ViTs over CNNs for regression tasks with satellite imagery, motivating future research to focus on applying ViT-based models for more accurate socio-economic indicator estimation.

### 6.4 FURTHER DISCUSSION

The ability to accurately measure poverty across a vast number of geolocations is crucial for understanding and addressing the disparities that exist in different regions. The extensive and high-quality poverty measurement is valuable for researchers and policymakers. It allows for the analysis of

poverty trends and the effectiveness of current policies, thereby facilitating more informed decision-making to reduce global poverty.

Traditional surveys, while rich in data, are limited by geographical and logistical constraints. Conducting extensive on-the-ground surveys is not only costly but also time-consuming—from data collection to processing and harmonization. In regions lacking detailed survey data, traditional methods like GPR or nearest-neighbor approaches are typically used to estimate interested variables. However, these methods can be unreliable, particularly when extrapolating data to locations far from surveyed areas or data with temporal dependencies, leading to high uncertainty.

On the other hand, satellite imagery, which was made widely available by organizations such as the ESA and the USGS, can be accessed from any geographic location. Recent advances in the field of computer vision have made it possible to infer meaningful information from this imagery, which can effectively improve poverty prediction. By demonstrating the capabilities of large vision models and satellite imagery in this context, we aim to inspire and encourage others in the field to further develop and refine these methods, thus driving changes in sociology research and policy making.

It is worth noting that our proposed method of combining satellite imagery with foundational vision models is not limited to only predicting child poverty. Various quantifiable variables, for example related to climate, conflicts, and pollution, may also be predicted by satellite data. This method provides a framework that could potentially be transferable for modeling many such variables. Future research could also attempt to adapt and apply this method to address other significant social and environmental science issues.

### 6.5 LIMITATION AND FUTURE DIRECTIONS

We present several limitations associated with our studies, along with suggestions for future directions. While DINOv2 achieves the best performance in our spatial benchmark, it has not yet exploited the full strength of multispectral remote sensing data. In the future, a multispectral self-distilled model for satellite usage could be investigated and compared with SatMAE when both are given the full spectrum of data. We highlight the difficulty of the temporal benchmark, suggesting that the model fails to capture temporal variation using satellite imagery alone. Future research could incorporate time-stamped data and explore time series methods for better forecasting performance. In addition, our estimation of child poverty is currently deterministic given the satellite imagery. Methods to quantify prediction uncertainty are paramount for the future development of this method, as they provide valuable information about spatially uncertain areas. This can guide policymakers and survey managers on where surveys need to be conducted. We note that while high-quality household survey data is expensive to acquire, it is an irreplaceable source of ground truth; machine learning can complement and enhance, but never replace, these datasets.

## 7 CONCLUSION

In conclusion, our study demonstrates the potential of combining satellite imagery with large vision models to estimate child poverty across spatial and temporal settings. We introduced a new dataset that pairs publicly accessible satellite images with detailed survey and child poverty data based on the Demographic and Health Surveys Program, covering 16 countries in Eastern and Southern Africa over the period 1997–2022. By benchmarking multiple models—including foundational vision models like MOSAIKS, DINOv2, and SatMAE—we assessed their performance in predicting child poverty. Our results show that advanced models using satellite imagery have the potential to outperform baseline methods, offering more accurate and generalizable poverty estimates. This work highlights the importance of integrating remote sensing data with machine learning techniques to address complex socioeconomic issues, providing a scalable and cost-effective approach for poverty estimation and policymaking.

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

# A APPENDIX

## A.1 DATASET DESCRIPTION

The KidSat dataset we present in this work includes both cluster-wise child poverty derived from the DHS data and the satellite imagery corresponding to each cluster. Due to the confidentiality of the survey data, DHS requires registration prior to accessing the data. We include detailed procedures for acquiring the satellite imagery and DHS data in our (anonymous) GitHub repository.

### A.1.1 IMAGERY STATISTICS

**Sentinel-2** is a mission from the European Space Agency (ESA), part of the Copernicus Programme. It consists of two satellites (Sentinel-2A and Sentinel-2B) and provides imagery in 13 spectral bands shown in Table 3.

| Band Name | Band Number | Central Wavelength (nm) | Resolution (m) |
|---|---|---|---|
| Coastal Aerosol | 1 | 443 | 60 |
| Blue | 2 | 494 | 10 |
| Green | 3 | 560 | 10 |
| Red | 4 | 665 | 10 |
| Red Edge 1 | 5 | 703 | 20 |
| Red Edge 2 | 6 | 740 | 20 |
| Red Edge 3 | 7 | 782 | 20 |
| NIR (Near Infrared) | 8 | 835 | 10 |
| NIR Narrow | 8A | 864 | 20 |
| Water Vapour | 9 | 945 | 60 |
| SWIR 1 | 11 | 1610 | 20 |
| SWIR 2 | 12 | 2190 | 20 |
| Cirrus | 10 | 1375 | 60 |

Table 3: Sentinel-2 Band Information

**Key Statistics for Sentinel-2:**

- **Spatial Resolution**: 10 m, 20 m, and 60 m depending on the band.
- **Temporal Resolution**: 5 days revisit time at the equator (with two satellites).
- **Spectral Range**: 13 spectral bands, ranging from visible light (Blue, Green, Red) to infrared (NIR and SWIR).
- **Coverage**: Global, with a swath width of 290 km.
- **Radiometric Resolution**: 12-bit data (values range from 0 to 4096).

**Landsat 5, 7, and 8** are parts of a long-running Earth observation mission managed by NASA and the U.S. Geological Survey (USGS). Here we provide band information in Table 4.

**Key Statistics for Landsat (Landsat 5, 7 & 8):**

- **Spatial Resolution**: 30 m for multispectral bands, 15 m for panchromatic, 100 m for thermal bands (resampled to 30 m).
- **Temporal Resolution**: 16 days revisit time.
- **Spectral Range**: 7 bands for Landsat 5, 8 bands for Landsat 7, and 11 bands for Landsat 8, spanning visible, infrared, and thermal wavelengths.
- **Coverage**: Global, with a swath width of 185 km.
- **Radiometric Resolution**: 16-bit data (values range from 0 to 65536).

We follow the conventional approach used in the Google Earth Engine for imagery normalization. To preprocess the Sentinel-2 imagery, we normalize the pixel values by scaling the original range

| Band Name | Landsat 5 | Landsat 7 | Landsat 8 | Wavelength (nm) |
|---|---|---|---|---|
| Coastal Aerosol | - | - | 1 | 433–453 |
| Blue | 1 | 1 | 2 | 450–520 |
| Green | 2 | 2 | 3 | 520–600 |
| Red | 3 | 3 | 4 | 630–670 |
| NIR (Near Infrared) | 4 | 4 | 5 | 850–880 |
| SWIR 1 | 5 | 5 | 6 | 1550–1750 |
| SWIR 2 | 7 | 7 | 7 | 2080–2350 |
| Panchromatic | - | 8 | 8 | 500–680 |
| Cirrus | - | - | 9 | 1360–1380 |
| Thermal Infrared 1 | 6 | 6 | 10 | 10400–12500 |
| Thermal Infrared 2 | - | - | 11 | 10400–12500 |

Table 4: Landsat 5, 7, and 8 Band Information

Table 5: This table categorizes various Demographic and Health Survey (DHS) variables by their respective child deprivation categories. The categories include Water, Sanitation, Nutrition, Health, Education, and Housing. Each category lists specific variables and their descriptions relevant to assessing child deprivation.

| Deprivation Category | Description | Variable |
|---|---|---|
| Water | Main drinking water source | hv201 |
| | Time to water source | hv204 |
| Sanitation | Type of toilet facility | hv205 |
| | Toilet sharing status | hv225 |
| Nutrition | Height-for-age z-score | hc70 |
| Health | Child received any vaccination | h10 |
| | DPT 1 vaccination | h3 |
| | DPT 2 vaccination | h5 |
| | DPT 3 vaccination | h7 |
| | Measles 1 vaccination | h9 |
| | Child had cough recently | h31 |
| | Current contraceptive method | v312 |
| Education | Highest education level in household | hv106 |
| | Educational attainment recoded | hv109 |
| | School attendance current year | hv121 |
| Housing | Sleeping rooms in household | hv216 |
| | Wealth index score | hv271 |

of 0 to 3000 to a range of 0 to 255. Values outside this range are clipped. This method preserves the relative intensity of the pixel values while adapting the data for image rendering. For Landsat imagery, the pixel values are normalized from 0 to 30000 to a range of 0 to 255. Similarly, values outside this range are clipped to ensure that they conform to the appropriate visualization range.

### A.1.2 CODING CHILD POVERTY

The severe_deprivation variable is used in this work to represent the percentage of children experiencing severe poverty for individual responses within the cluster. It is calculated by aggregating several indicators of severe deprivation across multiple dimensions such as housing, water, sanitation, nutrition, health, and education. The detailed definition of severe deprivation can be found in Table 1. Note that a child is classified in severe deprivation if they experience severe deprivation in any of the dimensions.

In addition, deprivation in each subcategory, as well as `moderate_deprivation`, is also included in the dataset. Further definitions can be found in the work by UNICEF (2021).

We present the histograms of the variable `severe_deprivation`, faceted by country, in Figure 4. The distributions of `severe_deprivation` vary significantly across countries. Most countries exhibit right-skewed distributions, with exceptions such as Malawi and Zimbabwe, which show left-skewed distributions. Additionally, some countries display Gaussian-like distributions (e.g., Rwanda), while others show U-shaped patterns (e.g., Tanzania). Given the variation in distribution across countries, spatial modeling for all of Eastern and Southern Africa poses a considerable challenge. For both optimization and policy-making purposes, country-specific modeling could improve the applicability and effectiveness of this approach.

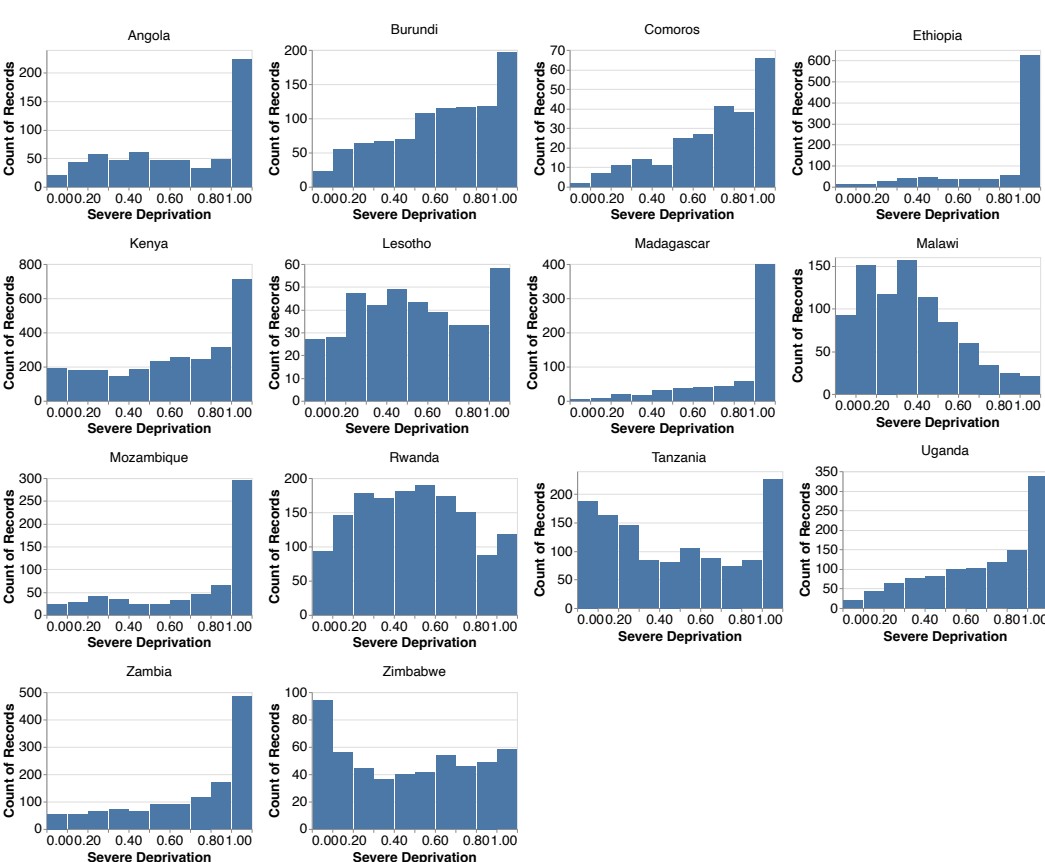

Figure 4: Histogram of the target severe deprivation variable, faceted by country.

Among all DHS variables used in child poverty calculation, we selected 17 variables, as presented in Table 5, as the prediction targets during model fine-tuning. Continuous variables were scaled to the range [0, 1], and categorical variables were expanded using one-hot encoding, where each category was represented by a binary indicator. This resulted in a 99-dimensional vector representing each cluster, based on the 17 selected DHS variables. We then used this vector to map satellite imagery for prediction and update as part of the model fine-tuning process.

## A.2 COMPUTE

As one of the heavy-lifting parts is loading images, a multi-core CPU ($\geq 8$) is recommended to optimize the data loading using multiple workers with the data loader. The training was done using Nvidia Tesla V100 GPUs for DINOv2 experiments and Nvidia L4 GPUs for SatMAE experiments. In particular, for DINOv2 experiments with Sentinel imagery, 32 GB of GPU memory is a hard requirement to process the full resolution of the input imagery.

## A.3 ADDITIONAL EXPERIMENTS

To demonstrate the effects of architectural changes, fine-tuning targets, and imagery resolution, we conducted a series of experiments, comparing models and configurations by altering each factor. The results are shown in Table 6 and the influence of these factors is discussed in the following three sections.

Table 6: Comparison of MAE on `severe_deprivation` across model architecture, fine-tuning target, imagery source, and input size in the spatial benchmark.

| Model | Fine-tune Target | Imagery Source | Input Size | MAE $\pm$ SE (Spatial) |
|---|---|---|---|---|
| DINOv2-ViT | Poverty Vector | Landsat | $336 \times 336$ | $0.2042 \pm 0.0015$ |
| DINOv2-ViT | Poverty Vector | Sentinel-2 | $994 \times 994$ | $0.1873 \pm 0.0022$ |
| DINOv2-ViT | Poverty Vector | Landsat | $224 \times 224$ | $0.2169 \pm 0.0017$ |
| DINOv2-ViT | Poverty Vector | Sentinel-2 | $224 \times 224$ | $0.2018 \pm 0.0028$ |
| DINOv2-ViT | Severe Deprivation | Landsat | $336 \times 336$ | $0.2114 \pm 0.0011$ |
| DINOv2-ViT | Severe Deprivation | Sentinel-2 | $994 \times 994$ | $0.1872 \pm 0.0021$ |
| DINO-ResNet50 | Poverty Vector | Landsat | $224 \times 224$ | $0.2401 \pm 0.0012$ |
| DINO-ResNet50 | Poverty Vector | Sentinel-2 | $224 \times 224$ | $0.2399 \pm 0.0015$ |
| SatMAE | Poverty Vector | Landsat | $224 \times 224$ | $0.2125 \pm 0.0019$ |
| SatMAE | Poverty Vector | Sentinel-2 | $224 \times 224$ | $0.2093 \pm 0.0039$ |

### A.3.1 ARCHITECTURE EFFECT: CNN VS. VIT

As a CNN-based model, the ResNet50 architecture requires the input size to be fixed and downsampled to $224 \times 224$, matching the input resolution used for SatMAE. In both the Landsat and Sentinel-2 experiments, we observe that DINOv2 with a ViT backbone and SatMAE with a transformer backbone outperform DINO with ResNet50. This highlights the architectural improvements, highlighting the superior performance of ViT in the child poverty estimation task.

### A.3.2 DINO RESIZE

To examine the effects of high-resolution input, we downsampled the input for DINOv2-ViT to $224 \times 224$. As shown in Table 6, compared to DINOv2 with full-resolution imagery, the performance decreases with downsampled inputs. However, the best-performing DINOv2-ViT model (0.2018) still outperforms the SatMAE model (0.2093) when using imagery of the same resolution.

### A.3.3 DIRECT OPTIMIZATION

We also conducted an experiment where the model was directly fine-tuned on the target variable, `severe_deprivation`, rather than the poverty vector expanded from the 17 DHS variables. We found that, when using Sentinel-2 imagery, directly optimizing for the target variable achieved comparable performance to fine-tuning with the poverty vector. However, in the experiment using Landsat imagery, direct optimization on the target variable led to worse performance. This may be because the poverty vector contains more comprehensive information that underpins the formulation of the severe deprivation variable, making the satellite features fine-tuned on this vector more robust and generalizable.

