# OpenReview forum: "KidSat: satellite imagery to map childhood poverty"
_ICLR.cc/2025/Conference — Submitted to ICLR 2025_

### Official Review · Reviewer_sp23 · 2024-10-31

**Soundness:** 2
**Presentation:** 3
**Contribution:** 1
**Rating:** 1
**Confidence:** 5

**Summary:**

This paper presents a new benchmark dataset called KidSat for predicting childhood poverty severity using satellite imagery from the Sentinel-2 and Landsat satellites. The prediction targets are constructed from DHS survey data in Eastern and Southern Africa spanning 1997-2022. The authors compared three models that process satellite imagery - MOSAIKS, DINO, and SatMAE - as well as two no-imagery baselines that rely only on the survey data itself - Gaussian Process Regression and Mean Regression - on the proposed benchmark.

**Strengths:**

- The paper gives sufficient background about the definition of childhood poverty and how it is measured
- The paper contributes a benchmark dataset that covers an application area of health/development that is not well covered by existing benchmarks
- Two simple baselines that use no satellite imagery are reported to ground the lower limit of expected performance

**Weaknesses:**

- The biggest limitation of this paper in my opinion is that it is unclear if this problem should be formulated as a satellite ML benchmark at all. Since the measures in the child poverty index are in most cases not directly observable from satellite data (especially at the resolution of Landsat and S2), it's unclear what features the authors expect to be learned. They even say on L385-386: "Satellite imagery is at best a proxy for multidimensional child poverty." Should we be prioritizing the use of ML to predict variables based on satellite data when that data is *at best* a proxy (at worst, not informative at all?)? What are the risks of the conclusions that can be drawn from such analyses?

- The results in Table 2 show that *some* useful information might be present in the satellite data because all of the imagery-based models do at least marginally better than the no-imagery baselines. However, it's not clear from the results if more sophisticated methods are needed to improve results, or if more sophisticated data are needed, since all of the models perform in a similar range that is not dramatically better than a no-imagery baseline.

- There is a lot of space in the text spent on redundant descriptions of the analysis, data, and discussion. Much of this space would have been better spent on additional experiments that would help better characterize the results and their utility, and evaluation with more modeling scenarios. For example:
  - The authors only tested self-supervised foundation models that ingest images. I think simpler baselines that use the satellite data but do not learn spatial correlations, like a random forest or other pixel-based methods, would be useful baselines to include to test the hypothesis that spatial structure is actually being learned.
  - The authors say that they only used RGB imagery to "directly compare" DINO with SatMAE. However, these methods are not directly comparable anyway - for example, different setups for the temporal benchmark. This does not seem like a sufficient reason not to test the multispectral versions of the data with SatMAE.
  - Results are reported as an average across the entire test dataset. Disaggregating these results by subgroup (e.g., country or the bins in Figure A.4) would give a more nuanced understanding of model performance and differences in performance. This would also be more useful for stakeholders to understand the utility (or lack thereof) of using the methods measured on this benchmark.

- Why was the SatMAE temporal model used if you did not include temporal information (replicated the same single-timestep input)? There is no temporal information being given to the model (except maybe the year of the data?).

- Questions about the dataset
  - The Sentinel-2 archive is only available since 2015/2016 but the survey data covers 1997-2022. Does this mean less data is used in the Sentinel-2 experiments?
  - For Landsat, data was drawn from Landsats 5, 7, and 8 to cover the survey period. There are some differences between these three datasets that could increase the heterogeneity of observations in the survey period, potentially a source of error that was not discussed in the paper. Did the authors used a harmonized version of these products?

**Questions:**

- Should we be prioritizing the use of ML to predict variables based on satellite data when that data is *at best* a proxy (at worst, not informative at all?)? What are the risks of the conclusions that can be drawn from such analyses?

- How do simpler ML baselines that do use satellite imagery inputs perform compared to the foundation models?

- How do the models perform on disaggregated versions of the test data, e.g. by country or by the bins in Figure 4.A?

- The Sentinel-2 archive is only available since 2015/2016 but the survey data covers 1997-2022. Does this mean less data is used in the Sentinel-2 experiments?

- For Landsat, data was drawn from Landsats 5, 7, and 8 to cover the survey period. There are some differences between these three datasets that could increase the heterogeneity of observations in the survey period, potentially a source of error that was not discussed in the paper. Did the authors used a harmonized version of these products?

- Why was the SatMAE temporal model used if you did not include temporal information (replicated the same single-timestep input)? There is no temporal information being given to the model (except maybe the year of the data?).

**Details Of Ethics Concerns:**

The biggest limitation of this paper in my opinion is that it is unclear if this problem should be formulated as a satellite ML benchmark at all. Since the measures in the child poverty index are in most cases not directly observable from satellite data (especially at the resolution of Landsat and S2), it's unclear what features the authors expect to be learned. They even say on L385-386: "Satellite imagery is at best a proxy for multidimensional child poverty." Should we be prioritizing the use of ML to predict variables based on satellite data when that data is *at best* a proxy (at worst, not informative at all?)? What are the risks of the conclusions that can be drawn from such analyses? The authors gave no discussion of the risks and potential negative impacts that could come from this.

---

### Official Review · Reviewer_rEvC · 2024-11-03

**Soundness:** 2
**Presentation:** 2
**Contribution:** 3
**Rating:** 3
**Confidence:** 4

**Summary:**

The paper proposes a dataset on child poverty featuring Sentinel-2 and Landsat imagery in Eastern and Southern Africa. Two dataset splits, a spatial and a temporal one, are proposed. The dataset is tested with several deep learning models such as MOSAIKS, Dinov2, and SatMAE, as well as two baseline approaches (mean regression and Gaussian process regression).

In general, the field benefits from challenging datasets. However, in this case, none of the factors constituting the target class, child poverty/prevalence of severe deprivation, are directly measurable from optical satellite imagery. Moreover, the dataset fails to incorporate key information such as nighttime light data that is typically used as a proxy for poverty. The paper also lacks visualizations that show examples of the satellite imagery, target value, and model outputs. Due to this and the fact that the spatial dataset split greatly simplifies the task, it is unclear whether models are capable of learning any information related to child poverty from this dataset.

**Strengths:**

- The dataset focuses on an important task in the Global South, while most existing remote sensing datasets were developed for the Global North.

- The paper includes two simple baselines that help to understand the dataset.

**Weaknesses:**

- The description of the satellite data is incomplete (I am aware that the satellite specifications are listed in the supplementary material but I am referring to the preprocessed satellite imagery used as model input). For example, key information such as the processing level, the selected spectral bands, and the spatial resolution of the satellite imagery are missing. This also applies to the abstract which does not provide any information on the satellite imagery.

- A literature review on relevant remote sensing studies mapping poverty is missing. Instead, the related work section cites several less relevant datasets such as xView2 or SpaceNet 7.

- Since nighttime light data is a key modality for poverty mapping, it is surprising that the authors did not include it. Therefore, I strongly suggest enriching the dataset with nighttime light data from, for example, SDGSAT-1 (ideal due to its high spatial resolution) or VIIRS.

- The visualizations are not useful for understanding the spatial patterns the models produce since they cover extensive areas. Therefore, I request some examples of model outputs over metropolitan areas of cities known to have deprived neighborhoods  (e.g., Maputo, Mozambique). This will lead to great insight into the capability of models trained on KidSat.

- For the spatial benchmark, I assume the clusters were randomly split into five folds. However, this split largely overlooks spatial autocorrelations and does not sufficiently test generalization ability across space (see Rolf et al., 2024). This limitation is also indicated by the performance of the Gaussian process regression baseline. Therefore, I recommend adding experiments for a region-based split.

**Questions:**

Have the authors considered using building data as input instead of satellite imagery? A good dataset for this could be the recent Open Buildings 2.5D Temporal Dataset, which provides information on building presence, fractional building count, and building height.

---

### Official Review · Reviewer_YGsN · 2024-11-03

**Soundness:** 2
**Presentation:** 2
**Contribution:** 1
**Rating:** 1
**Confidence:** 4

**Summary:**

The paper outlines a methodology to map poverty (i.e., severe deprivation) across southern Africa by encoding satellite images into feature representations with pre-trained models (Dino, SatMAE) , random kitchen sinks (MOSAIKS) and performing a linear regression on the target variable "severe deprivation". A baseline is the GP regression given the label data.

While the problem is highly relevant and impacting, the implications for the Machine Learning community remain unclear. In particular, the discussion remains focused on poverty mapping and does not investigate an underlying methodological question of "which pre-training or feature representations are particularly useful for certain downstream tasks (like poverty mapping)". Currently, tested methods are used as is, and their feature vectors have not been investigated in depth.

Hence, i believe that this paper would rather fit in an applied journal where reviewers can better evaluate the societal impact of poverty mapping with existing foundation models.

**Strengths:**

* Impacting problem and extensive dataset.

**Weaknesses:**

* methodological questions are not investigated in-depth. E.g., what impact does the pre-training algorithm have on downstream performance in general. Are (reconstruction-based) MAE features better than Dino (joint-embedding based) features (see for instance, Shekhar et al., 2023 for a comparison here) in general for poverty mapping? Do takeaways in that direction align with existing papers?

Shekhar, S., Bordes, F., Vincent, P., & Morcos, A. (2023). Objectives matter: Understanding the impact of self-supervised objectives on vision transformer representations. arXiv preprint arXiv:2304.13089.

**Questions:**

* What implications/impact does this paper have to the machine learning community beyond poverty mapping?
* Why was the focus on severe_deprivation and not other poverty indicators in the poverty vector? Is severe_deprivation a particularly well-suited variables to test different foundation models?

---

### Official Review · Reviewer_REgw · 2024-11-05

**Soundness:** 2
**Presentation:** 1
**Contribution:** 1
**Rating:** 3
**Confidence:** 5

**Summary:**

The paper "KidSat: Satellite Imagery to Map Childhood Poverty" presents a dataset and benchmarking framework aimed at improving the analysis of child poverty indicators in Eastern and Southern Africa through satellite imagery. This research provides a dataset consisting of 33,608 images covering 16 countries from 1997 to 2022, paired with high-quality survey data on multidimensional child poverty factors such as housing, sanitation, water, nutrition, education, and health. By leveraging geocoded Demographic and Health Surveys data, the authors have created a resource for the social science and global health community, enabling detailed analyses across both spatial and temporal dimensions.
This work appears notable for its systematic comparison of feature representations, ranging from low-level satellite models like MOSAIKS to state-of-the-art deep learning foundation models, including DINOv2 and SatMAE. The authors assess each model's capacity for spatial generalization across different locations and temporal generalization beyond the training years, providing a framework to evaluate model efficacy across varied contexts.

**Strengths:**

Originality
The "KidSat" paper presents a dataset that combines high-resolution satellite imagery with multidimensional child poverty data from Demographic and Health Surveys (DHS) across 16 countries in Eastern and Southern Africa. This dataset applies satellite imagery to estimate socioeconomic indicators related to child poverty, focusing on UNICEF’s six poverty dimensions—housing, water, sanitation, nutrition, health, and education. The dataset introduces a method for analyzing child-specific poverty indicators rather than general economic metrics.

Quality
The paper follows a methodical approach, providing reproducible steps for dataset construction and benchmark testing. The authors compare a range of models, including Gaussian Process Regression, MOSAIKS, DINOv2, and SatMAE, to assess performance on spatial and temporal generalization tasks. Fine-tuning, spatial and temporal benchmarks, and cross-validation are used to evaluate each model’s generalization capability.

Clarity
The paper is structured to clearly present the models, dataset, and evaluation metrics. The benchmark results, including error metrics for each model type, allow for straightforward comparisons of model performance in child poverty prediction. Visual aids, such as the heatmap of survey cluster locations and summary tables, provide concise representations of key findings.

Significance
This research provides a benchmark for child poverty estimation using satellite imagery, offering potential applications for data-informed policy and social science research in regions lacking detailed survey data. The dataset and benchmark may support further studies in poverty estimation using satellite data. Given the global availability of satellite imagery, this framework could be adapted to analyze other social or environmental indicators.

**Weaknesses:**

1. Limited Exploitation of Multispectral Data
While the "KidSat" dataset includes multispectral satellite imagery from sources like Sentinel-2, the study does not fully utilize the additional spectral bands beyond RGB. The paper mentions that DINOv2 relies solely on RGB bands, potentially limiting the model’s ability to capture information relevant to socioeconomic indicators, such as vegetation health or water bodies. Future work could include experiments with multispectral configurations or foundation models designed to leverage the entire spectral range available in Sentinel and Landsat imagery.

2. Constraints on Temporal Prediction Capability
The temporal benchmark reveals challenges in predicting poverty indicators in years beyond the training period (2020–2022). This suggests that the current models may be overfitting to historical data patterns and struggle to generalize to newer datasets. Incorporating temporal encoding or exploring time-series models may improve the framework’s capability to handle time-sensitive socioeconomic changes. Furthermore, training models with time-stamped data could aid in detecting trends, which would be valuable for forecasting.

3. Fixed Input Size and Resizing Effects on Model Performance
The study limits DINOv2 and SatMAE’s performance by requiring the imagery to be resized to a specific input size (224×224 for certain configurations). Resizing can lead to information loss, particularly when working with high-resolution satellite images. Evaluating models with variable or adaptive input sizes, or employing patch-based processing for high-resolution images, might preserve more spatial detail and enhance prediction accuracy.

4. Absence of Uncertainty Estimation
The current approach is deterministic and does not quantify uncertainty in model predictions, which is crucial for policymakers who may rely on this data to inform decisions. Including uncertainty metrics, such as confidence intervals or probabilistic models, would provide a more comprehensive assessment of model reliability, particularly in areas with sparse data coverage.

5. Limited Comparison with Non-ViT Architectures
Although the study highlights the use of Vision Transformers (ViTs), the experiments with alternative architectures, like CNN-based models, are limited and do not include any recent transformer alternatives. Comparing ViTs with other state-of-the-art architectures, particularly those optimized for spatial analysis, would help clarify the advantages and limitations of ViTs in the context of satellite imagery-based poverty estimation.

6. Dataset Coverage and Geographic Bias
The dataset, while extensive, is geographically focused on Eastern and Southern Africa. This focus may limit the applicability of the benchmark models to other regions with different socioeconomic and environmental conditions. An expanded dataset that includes diverse regions globally could help generalize the findings and provide a more comprehensive tool for poverty estimation.

7. Lack of Direct Validation with Ground Truth
The study’s reliance on DHS survey data as ground truth is practical, but there is limited discussion on validating model predictions with additional sources or cross-referencing with other poverty metrics. Incorporating data from other surveys, censuses, or economic indicators could enhance model robustness and provide a broader validation framework.

**Questions:**

1. Utilization of Multispectral Data
Could you provide insights into why the multispectral bands of Sentinel-2 were not fully leveraged in the models? Given that some indicators relevant to child poverty (e.g., vegetation indices for assessing agricultural productivity) could benefit from multispectral analysis, do you have plans to experiment with models that use the full spectral range? If so, which models do you anticipate could best handle this data?

2. Temporal Benchmarking and Forecasting Capability
The temporal benchmark indicates some difficulty in generalizing to future years. Could you clarify whether you attempted any temporal encoding techniques or time-series models in preliminary tests? Additionally, do you believe that training with time-stamped data would improve temporal generalization, and if so, what data sources might you consider?

3. Adaptive Image Resolution
In the current experiments, fixed input sizes (e.g., 224×224) are required for some models, which could result in a loss of spatial detail when processing high-resolution satellite images. Have you considered using models with patch-based processing or adaptive resolutions to maintain image fidelity? If so, could you share any initial observations or challenges you faced with these approaches?

4. Uncertainty Estimation
Since the predictions are deterministic, how do you envision integrating uncertainty estimation into the model outputs? For example, would probabilistic models or Bayesian approaches fit into your pipeline, and are there particular techniques you see as most promising for generating uncertainty estimates?

5. Geographic Generalizability
Given that the dataset focuses on Eastern and Southern Africa, have you considered testing on data from other regions, or could you discuss any plans to expand the dataset to other continents? Understanding how your models perform across diverse socioeconomic and environmental contexts would enhance the generalizability of this framework.

6. Validation with Alternative Ground Truth Sources
The paper uses DHS survey data as the primary ground truth. Could you elaborate on any efforts or future plans to validate predictions with additional poverty indicators or socioeconomic datasets? Are there other data sources you would consider integrating to broaden the validation framework?

7. Vision Transformer (ViT) Model Comparison
You’ve highlighted the performance of ViTs in this task. Could you clarify whether you have plans to compare ViTs with any recent transformer-based architectures optimized for spatial analysis? Additionally, did you consider using transformer architectures specifically pre-trained on remote sensing tasks, and if so, how did they perform?

---

### Meta-Review · Area_Chair_mWwL · 2024-12-14

**Metareview:**

Dear authors,

Thank you for submitting the draft. The reviewers' rankings indicated that the draft is not ready for publication at this stage.

The draft proposes a new dataset to predict child poverty from satellite imagery.  Results are presented for both locations outside the training set, and "on data beyond the training years". Reviewers gave detailed comments, especially REgw, sp23, and rEvC. rEvC raised concern about missing relevant studies on mapping poverty. A concern raised by REgw was "Constraints on Temporal Prediction Capability" that is whether it can predict poverty beyond the training time of 2020-2022. sp23 raised a question should this problem be formulated as a satellite ML benchmark at all, this indicates that the draft lacks clarity to explain the need and promise of their methodology? The authors did not provide feedback.

We hope comments by reviewers will help improve the draft.

regards

AC

**Additional Comments On Reviewer Discussion:**

All reviewers agree draft should not be accepted at this stage.

---

### Decision · Program_Chairs · 2025-01-22

Reject